# Essential Role of Host Double-Stranded DNA Released from Dying Cells by Cationic Liposomes for Mucosal Adjuvanticity

**DOI:** 10.3390/vaccines8010008

**Published:** 2019-12-27

**Authors:** Rui Tada, Akihiro Ohshima, Yuya Tanazawa, Akari Ohmi, Saeko Takahashi, Hiroshi Kiyono, Jun Kunisawa, Yukihiko Aramaki, Yoichi Negishi

**Affiliations:** 1Department of Drug Delivery and Molecular Biopharmaceutics, School of Pharmacy, Tokyo University of Pharmacy and Life Sciences, Tokyo 192-0392, Japanaramaki@toyaku.ac.jp (Y.A.); negishi@toyaku.ac.jp (Y.N.); 2Division of Mucosal Immunology and International Research and Development Center for Mucosal Vaccines, Department of Microbiology and Immunology, The Institute of Medical Science, The University of Tokyo, Tokyo 108-8639, Japan; kiyono@ims.u-tokyo.ac.jp (H.K.); kunisawa@nibiohn.go.jp (J.K.); 3Laboratory of Vaccine Materials and Laboratory of Gut Environmental System, National Institutes of Biomedical Innovation, Health and Nutrition (NIBIOHN), Osaka 567-0085, Japan

**Keywords:** cationic liposome, DAMPs, dsDNA, dying cells, mucosal adjuvant

## Abstract

Infectious disease remains a substantial cause of death. To overcome this issue, mucosal vaccine systems are considered to be a promising strategy. Yet, none are approved for clinical use, except for live-attenuated mucosal vaccines, mainly owing to the lack of effective and safe systems to induce antigen-specific immune responses in the mucosal compartment. We have reported that intranasal vaccination of an antigenic protein, with cationic liposomes composed of 1,2-dioleoyl-3-trimethylammonium-propane and 3β-[N-(N′,N′-dimethylaminoethane)-carbamoyl], induced antigen-specific mucosal and systemic antibody responses in mice. However, precise molecular mechanism(s) underlying the mucosal adjuvant effects of cationic liposomes remain to be uncovered. Here, we show that a host double-stranded DNA (dsDNA), released at the site of cationic liposome injection, plays an essential role for the mucosal adjuvanticity of the cationic liposome. Namely, we found that nasal administration of the cationic liposomes induced localized cell death, at the site of injection, resulting in extracellular leakage of host dsDNA. Additionally, *in vivo* DNase I treatment markedly impaired OVA-specific mucosal and systemic antibody production exerted by cationic liposomes. Our report reveals that host dsDNA, released from local dying cells, acts as a damage-associated molecular pattern that mediates the mucosal adjuvant activity of cationic liposomes.

## 1. Introduction

Infectious diseases are responsible for a substantial number of deaths, despite developments in modern medicine. To date, live-attenuated and/or inactivated vaccines are widely utilized for human clinical use, though they may cause detrimental effects especially in newborns, elders, or immunodeficient patients [1,2]. Therefore, clinics urgently need more safe and effective vaccine strategies by means of split and/or subunit vaccines. Emerging mucosal vaccines are considered to be the most promising way to prevent death caused by pathogens because, unlike parenteral vaccinations, mucosal vaccines are capable of inducing specific immune responses to pathogens in mucosal compartments, the common port of entry or site of colonization for most microbial pathogens. Yet, no mucosal vaccines have been approved for clinical use except for live-attenuated mucosal vaccines [3,4,5,6]. Thus, the development of mucosal adjuvant is crucial, owing to inherently poor immunogenicity of antigens at mucosal sites, such as the nasal route [7].

Adjuvants are commonly used to enhance antigen-specific immune responses against antigens, which are co-administered as part of vaccine formulations, thereby improving not only vaccine efficiency, but also long-term immunological memory [8,9]. As the mucosal surface is regularly provoked by an abundance of antigens and microbiota, the mucosal immune system is tightly regulated in a tolerant manner [10]. Therefore, an appropriate adjuvant would be required to effectively induce an antigen-specific immune response when administered via mucosal routes. However, alum, which is one of the most used adjuvants for parenteral vaccines, is known to poorly induce mucosal immune responses to the antigen, indicating that the development of effective and safe mucosal adjuvants is still needed [11].

We have recently elucidated that intranasal vaccination of an antigenic protein with cationic liposomes composed of 1,2-dioleoyl-3-trimethylammonium-propane (DOTAP) and 3β-[N-(N′,N′-dimethylaminoethane)-carbamoyl] (DC-chol) (DOTAP/DC-chol liposomes) induced antigen-specific mucosal and systemic antibody responses in mice [12,13,14]. Namely, the DOTAP/DC-chol liposome acts as a mucosal adjuvant for nasal vaccine formulations, especially for infectious diseases [15]. However, precise molecular mechanism(s) behind the mucosal adjuvant effects of cationic liposomes are still not clarified. Cationic liposomes, at least in part, enhance antigen uptake by antigen-presenting cells (APCs), such as dendritic cells (DCs), both *in vitro* and *in vivo* [12,16], which indicates that the DOTAP/DC-chol liposome acts as an antigen delivery carrier. In contrast, recent research has highlighted that adjuvants activate the innate immune system through recognition of damage-associated molecular patterns (DAMPs) released from dying cells. Yet, it has long been believed that adjuvants create a depot effect through sustained antigen exposure or facilitate antigen uptake by APCs. Namely, the released DAMPs, such as high-mobility group box 1 (HMGB1), uric acid crystals, adenosine triphosphate (ATP), and nucleic acids, can alert the host innate immune system via the activation of various signaling pathways, including those mediated by host pattern-recognition receptors (PRRs) [17]. For instance, the most commonly used adjuvant for human vaccines, alum, acts thorough release of host double-stranded DNA (dsDNA) from dying cells at the site of injection. As dsDNA is normally located inside the cell, it thus acts as a DAMP and activates the innate immune system [18,19].

To explore the mechanism(s) of action of the cationic liposomes, we hypothesized that DAMPs released at the immunization site of DOTAP/DC-chol liposomes might contribute to the mucosal adjuvant effects of cationic liposomes. This work is motivated by our long reported evidence that cationic liposomes can induce mitochondrial oxidative stress-mediated cell death of a wide range of cells *in vitro* [20,21,22,23]. In the present study, we investigated the involvement of DAMPs in antigen-specific antibody responses induced by DOTAP/DC-chol liposomes in both mucosal and systemic compartments in mice. Herein, we observed that released host dsDNA, at the site of DOTAP/DC-chol liposome injection, plays an essential role for the adjuvanticity of cationic liposomes.

## 2. Materials and Methods 

### 2.1. Mice

We purchased female BALB/cCrSlc mice (six weeks of age) from Japan SLC (Shizuoka, Japan). All mice were housed in a specific pathogen-free facility. Aged match-groups of mice were used at 7–11 weeks of age in all experiments. All experimental protocols were approved in advance by the Committee for Laboratory Animal Experiments at the Tokyo University of Pharmacy and Life Sciences (P15–33, P16–12, P17–26, P18–71, and P19–58).

### 2.2. Materials

DOTAP and DC-chol were purchased from Avanti Polar Lipids (Alabaster, AL, USA). Low endotoxin (<1 EU/mg) egg white ovalbumin (OVA) and endotoxin free phosphate-buffered saline (PBS) were obtained from FUJIFILM Wako Pure Chemical Industries (Osaka, Japan). Endotoxin free Hanks’s balanced salt solution containing Ca^2+^ and Mg^2+^ (HBSS) was purchased from Nacalai Tesque (Kyoto, Japan).

### 2.3. Liposome Preparation

The cationic liposomes used in this study were prepared as previously described [12,13,14,15,16]. In short, 10 μmol of total lipids (DOTAP/DC-chol at 1:1 mol ratios) was evaporated to dryness in a glass tube and desiccated for 1 h *in vacuo*. The obtained lipid films were hydrated by the addition of 250 μL of PBS and then vortexed for 5 min. The multilamellar vesicles were extruded 10 times by passing through a 100 nm pore polycarbonate membrane (ADVANTEC, Tokyo, Japan) and sterilized by 0.2 μm of cellulose acetate membrane filters (IWAKI, Tokyo, Japan). The particle size and ζ-potential were verified with NICOMP 380 ZLS (Particle Sizing Systems; Port Richey, FL, USA).

### 2.4. Immunization and Sampling Schedule

Mice were anesthetized via intraperitoneal injection of 0.2 mL of a mixture containing 0.75 mg/kg of medetomidine, 4 mg/kg of midazolam, and 5 mg/kg of butorphanol tartrate. They were then immunized intranasally with (1) PBS, (2) OVA (2.5 μg/mouse) alone, or (3) OVA (2.5 μg/mouse) plus liposomes (400 nmol/mouse). Each group of mice was immunized once weekly for three weeks. Blood samples were collected, allowed to clot at 25 °C for 30 min, and incubated at 4 °C for 60 min. Serum was separated by centrifugation at 1200× *g* for 30 min. Nasal wash fluid, bronchoalveolar lavage fluid (BALF), and vaginal wash fluid were collected by 200 µL, 1 mL, and 100 µL of cold PBS, respectively [24,25]. All samples were stored at −80 °C until enzyme-linked immunosorbent assay (ELISA) analysis.

### 2.5. ELISA Assay for the Detection of OVA-Specific Antibody Titers

A Nunc MaxiSorp 96-well plate (ThermoFisher Scientific, MA, USA) was coated with 125 ng of OVA in 0.1 M carbonate buffer (pH 9.5) and incubated overnight at 4 °C. The plate was then washed with PBS containing 0.05% Tween 20 (PBST) and blocked with 1% bovine serum albumin (BSA; FUJIFILM Wako Pure Chemical Industries) containing PBST (BPBST) at 37 °C for 60 min. The plate was washed and incubated with serum or mucosal fluid samples overnight at 4°C. The PBST washed plate was then treated with peroxidase-conjugated anti-mouse IgA (Cat. No. 1040-05), IgG (Cat. No. 1031-05), IgG1 (Cat. No. 1070-05), or IgG2a (Cat. No. 1080-05) secondary antibody (SouthernBiotech, Birmingham, AL, USA) in BPBST, and developed using a tetramethylbenzidine substrate system (KPL, Milford, MA, USA). Color development was terminated using 1 N phosphoric acid and the optical density was measured at 450 nm with 650 nm of reference using a Synergy HTX Multi-Mode Microplate Reader (BioTek Instruments, Inc., Whiting, VT, USA) [12]. The endpoint titers were calculated as the reciprocal of the last dilution reaching a cut-off value set to twice the mean optical density of a negative control [26,27].

### 2.6. DsDNA Quantification in Nasal Fluids

Nasal wash samples were collected at 1 to 16 h after intranasal administration of liposomes (400 nmol/mouse) by flushing with 100 µL of cold PBS. After centrifugation to remove cells/debris, the amounts of dsDNA in the nasal wash samples were measured by a Fluorescent DNA Quantitation Kit (Bio-Rad Laboratories, Hercules, CA, USA) according to the manufacturer’s instructions using a Synergy HTX Multi-Mode Microplate Reader (BioTek Instruments, Inc.).

### 2.7. Flow Cytometric Assessment of Dying Cells

Female BALB/cCrSlc mice (seven weeks of age) were administrated with PBS (vehicle) or liposomes (400 nmol/mouse) intranasally. Six hours after administration, single cells from nasal tissues were isolated by mechanical dissociation using stainless-steel mesh, followed by passing through 70 µm nylon mesh. The isolated cells were then treated with Red Blood Cell (RBC) lysing buffer (BioLegend, San Diego, CA, USA) to lyse red blood cells. After washing the cells with PBS, dead cells in the preparation were stained with 7-amino-actinomycin D (7-AAD) viability staining solution (BioLegend), according to the manufacturer protocol, and then analyzed using a FACSCanto instrument (BD Biosciences, San Jose, CA, USA).

### 2.8. Preparation of Murine Genomic DNA

Murine genomic DNA was isolated from spleen excised from female BALB/cCrSlc mice using a NucleoSpin Tissue kit (Takara Bio, Tokyo, Japan) according to the manufacturer’s protocol. The prepared genomic DNA was verified by agarose gel electrophoresis followed by visualization with GelRed nucleic acid gel stain (FUJIFILM Wako Pure Chemical Industries).

### 2.9. Mucosal Adjuvant Effects of Nasally Administered Genomic DNA

Mice were immunized intranasally with (1) PBS, (2) OVA (2.5 µg/mouse) alone, or (3) OVA (2.5 µg/mouse) plus genomic DNA (50 ng/mouse). Each group of mice was immunized once weekly for three weeks. A week after the last immunization, blood and mucosal samples were collected as described above.

### 2.10. In Vivo DNase I Treatment

For liposome experiments, mice were immunized intranasally with OVA alone or OVA in combination with liposomes once weekly for three weeks. Three and eighteen hours after each immunization, mice nasally received 200 IU of RNase-free recombinant DNase I (Sigma-Aldrich, St. Louis, MO, USA) in 10 µL of Hanks' Balanced Salt Solution (HBSS) containing Ca^2+^ and Mg^2+^ [19]. A week after the last immunization, blood and mucosal samples were collected as described above.

For genomic DNA experiments, mice were immunized intranasally with OVA (2.5 µg/mouse) alone, OVA (2.5 µg/mouse) plus genomic DNA (50 ng/mouse), OVA (2.5 µg/mouse) plus genomic DNA pre-treated with RNase-free recombinant DNase I (200 IU) for 30 min (50 ng/mouse), or OVA (2.5 µg/mouse) plus genomic DNA (50 ng/mouse), followed by the immediate treatment of nasal DNase I (200 IU) once weekly for three weeks. A week after the last immunization, blood and mucosal samples were collected as described above.

### 2.11. Statistical Analysis

Statistical differences were calculated using the Kruskal–Wallis with Dunn’s post-hoc test, Mann–Whitney U test, or *t*-test with Welch’s correction. *p*-values < 0.05 were considered significant.

## 3. Results

### 3.1. Cationic Liposomes Enhance Antigen-Specific Mucosal and Systemic Antibody Responses

We first tested whether intranasal immunization of cationic liposomes in combination with OVA induce OVA-specific antibody responses in both systemic and mucosal compartments, including nasal, bronchoalveolar, and vaginal lavages. As shown in Figure 1 and Figure 2, intranasal vaccination of cationic liposomes and OVA promoted the production of OVA-specific nasal IgA, BALF IgA, BALF IgG, and vaginal IgA (endpoint titer with median value: 103.3, 143.0, 8579, and 1915, respectively) when compared with mice immunized with OVA alone. Furthermore, nasally administered cationic liposomes exerted marked OVA-specific IgG, IgG1, and IgG2a responses in serum (endpoint titer with median value: 3.4 × 10^6^, 1.3 × 10^7^, and 6740, respectively) compared with mice administered OVA alone. Taken together, these data demonstrated that cationic liposomes act as a mucosa adjuvant in mice when administered via the nasal route, as expected [12].

### 3.2. Cationic Liposome Provokes Cell Death Followed by the Release of Host DsDNA at the Site of Injection

While the data have shown that cationic liposomes would be a favorable mucosal adjuvant, their mechanism of action is still unknown. Recent research has highlighted the importance of DAMPs released by dying host cells for the adjuvant activity of several molecules, such as alum, MF59, and Endocine [18,19,28,29,30]. In addition, it is well known that the cationic liposomes generally show cellular toxicity to a wide range of the cells *in vitro* [20,31,32,33]. Considering this evidence, we hypothesized that administration of cationic liposomes induces localized cell death and subsequent release of DAMPs, such as host dsDNA from dying cells, which in turn activates acquired immune responses toward antigenic proteins. To test this hypothesis, we examined extracellular dsDNA concentrations from the nasal cavity of mice that received DOTAP/DC-chol liposomes intranasally. Figure 3A shows that nasal administration of cationic liposomes induces leakage of host dsDNA into the nasal cavity at 6 h after treatment with gradual increase over time. Leakage of dsDNA was correlated with the cell death at the site of injection of the cationic liposomes (Figure 3B). Considering these results, the release of dsDNA from localized dying cells, induced by the cationic liposomes, might be involved with the activity of the mucosal adjuvant.

### 3.3. Host DsDNA Exerts Antigen-Specific Systemic and Mucosal Immune Responses

Extracellular host dsDNA is reported to trigger immune responses followed by induction of antigen-specific antibody responses. Thus, to investigate whether host dsDNA released by intranasal vaccination of cationic liposomes is involved in the mucosal adjuvant effects [34,35], we tested whether intranasal administration of dsDNA with OVA promote OVA-specific antibody production in mice (Figure 4). As expected, intranasal injection of OVA mixed with prepared mouse genomic DNA remarkably exerted OVA-specific mucosal nasal IgA, BALF IgA, BALF IgG, and vaginal IgA (endpoint titer with median value: 16.44, 3.280, 4463, and 61.00, respectively) when compared with mice immunized with OVA alone. Furthermore, nasally administered genomic DNA also exerted marked OVA-specific IgG and IgG1 responses in serum (endpoint titer with median value: 8.7 × 10^5^ and 1.8 × 10^6^, respectively) when compared with mice administered OVA alone. On the contrary, serum IgG2a production was not significantly induced in this experimental condition, indicating that host dsDNA may act as a mucosal adjuvant by boosting Th2-polarized humoral immune responses similar to those induced by the cationic liposome. Next, to rule out the presence of contaminating substances, such as endotoxin, in the prepared mouse genomic DNA, we investigated whether DNase I, an endo-deoxyribonuclease that destroys genomic DNA, abolished the mucosal adjuvant effects. We observed that treatment of DNase I almost completely abrogated OVA-specific antibody responses in both mucosal and systemic compartments, clearly demonstrating that mouse genomic DNA truly enhanced antigen-specific antibody responses as a mucosal adjuvant, maybe through triggering the innate immune system via recognition by host PRRs (Figure 5).

### 3.4. Mucosal Adjuvant Effect of Cationic Liposomes Relies upon Released Host DsDNA

Our described data implied that intranasal administration of cationic liposomes provoked host dsDNA release in the nasal cavity of mice by inducing cell death at the site of administration. In addition, mouse genomic DNA itself acts as a mucosal adjuvant in mice. Combining these results, we explored the hypothesis that the mucosal adjuvant effects of cationic liposomes is mediated by host dsDNA released from dying local cells induced by cationic liposomes. In order to evaluate this, we examined the effect of *in vivo* DNase I treatment on the mucosal adjuvant activities of cationic liposomes assessed by the production of OVA-specific antibody in mice immunized nasally with OVA in combination with DOTAP/DC-chol liposomes. As presented (Figure 6), *in vivo* DNase I treatment markedly impaired OVA-specific mucosal and systemic antibody production. Collectively, this study revealed that host dsDNA released from local dying cells acts as a DAMP that mediates the mucosal adjuvant activity of cationic liposomes.

## 4. Discussion

Although mucosal vaccines have long been considered a predominant strategy in overcoming fatal infectious diseases caused by pathogenic microbes, only a few live-attenuated and inactivated mucosal vaccines are utilized in the clinic. Specifically, no split and/or subunit type mucosal vaccines have been licensed for human use [36]. Hence, we investigated the use of mucosal adjuvants, for the formulation of mucosal vaccines, to enhance an antigen-specific immune response in mucosa, which is crucial for developing effective split and/or subunit mucosal vaccines. In the course of our study, we found that cationic liposomes composed of DOTAP and DC-chol act as a potent mucosal adjuvant, resulting in both systemic and mucosal antigen-specific antibody responses in mice, when nasally co-immunized with an antigenic protein [12,15]. However, precise molecular mechanism(s) underlying the mucosal adjuvant effects of the cationic liposomes are still undetermined. In this study, we demonstrated that cationic liposomes induce cell death at the site of injection, followed by leakage of host dsDNA into the nasal cavity. Our data also show that genomic DNA extracted from mice acts as a mucosal adjuvant. Moreover, *in vivo* DNase I treatment weakened the mucosal adjuvant activities of both cationic liposomes and host dsDNA. On the basis of these findings, the cationic liposome appears to exert mucosal adjuvant activity through the release of host dsDNA from local dying cells. Released dsDNA acts as DAMPs induced by the action of the cationic liposomes in mice when administered via the nasal route.

Intranasal immunization with OVA and DOTAP/DC-chol liposomes exhibited antigen-specific antibody responses in both mucosal and systemic compartments, as expected (Figure 1 and Figure 2). Even though we observed a very low background signal in the current experiments, one notable limitation of this study might be the use of BSA as a blocking agent in the ELISA performed to detect anti-OVA antibodies. Although BSA has previously been used as a blocking agent in ELISAs to detect anti-OVA antibodies [37,38], it has been reported that OVA and BSA possess immunologically similar epitopes, and the antibodies produced against OVA might thus cross-react with BSA, especially in polyclonal antisera derived from rabbits [39]. To explore mechanism(s) behind this activity, we tested the involvement of DAMPs in the mucosal adjuvant effects, because cationic liposomes are known to be cytotoxic *in vitro* [20,21,22,23,31,32,33]. While little is known about the toxicity of cationic liposomes *in vivo*, Wegmann et al. reported that polyethyleneimine (PEI), a cationic polymer, induces cell death *in vivo*, followed by the release of host dsDNA [40]. Thus, we first examined the percentage of dead cells at the site of administration of nasal cationic liposomes. Indeed, significant abundance of dead cells was observed on nasal mucosa after intranasal injection of the cationic liposomes (Figure 3B). Next, to examine host dsDNA release *in vivo*, we intranasally administered cationic liposomes to mice and quantified release of host dsDNA in nasal lavage. It was reported that intraperitoneally administered alum, as well as PEI, induced release of high concentrations of dsDNA into peritoneal lavage. Similarly, intranasally administered cationic liposomes exerted the release of host dsDNA into the nasal cavity (Figure 3A). These data persuaded us to investigate whether the treatment of DNase I, an endonuclease, reduces the mucosal adjuvant effects of cationic liposomes by degrading released host dsDNA, thus suppressing subsequent mucosal immune responses to the antigens. We observed that DNase I treatment significantly suppressed antigen-specific antibody responses induced by cationic liposomes (Figure 6), indicating that host dsDNA released from dying cells might be involved in the mucosal adjuvant effects of cationic liposomes. We further investigated whether host dsDNA itself possesses mucosal adjuvant effects in mice. Figure 4 and Figure 5 demonstrate that intranasal immunization of host dsDNA, in combination with OVA, remarkably exerted antigen-specific antibody responses both systemically and in mucosa. Cumulatively, host dsDNA released from dying cells, induced by cationic liposomes, has an essential role in the mucosal adjuvant activities of cationic liposomes in mice. In the present study, we used RNase- and protease-free DNase I for our experiments because Laura et al. reported that some commercial DNase I preparations are contaminated with proteases that confound the conclusion of the involvement of host dsDNA in the mechanisms underlying the adjuvanticity of alum [41]. Therefore, further experiments that use knock-out mice lacking various DNA sensors would be needed to confirm the involvement of host dsDNA in the mucosal adjuvanticity of cationic liposomes.

The present study shows that extracellular host dsDNA acts as DAMPs and is indispensable for the adjuvanticity of cationic liposomes. However, it remains unclear how host dsDNA activates antigen-specific antibody responses in both mucosal and systemic compartments. Several studies have demonstrated that the sensing of nucleic acids as PAMPs or DAMPs by host PRRs effectively stimulate host immunity induced by live attenuated vaccines, alum, and DNA vaccines [42]. In addition, antigen-specific antibody responses, following vaccination with the water-soluble polysaccharide chitosan, were reported to be dependent on both the cytoplasmic DNA sensor cGAS and STING [43,44]. This suggests the possible contribution of PPRs, including DNA sensors, to the mucosal adjuvanticity of cationic liposomes. We are currently exploring whether we can identify not only the receptors responsible for those activities, but also subsequent immunological events.

In this study, we clearly demonstrated that extracellular host dsDNA released from dying cells in nasal mucosa induced by cationic liposome injection plays a pivotal role for the mucosal adjuvanticity of cationic liposome. While it is unknown how host dsDNA, released in response to adjuvants intranasally co-administered with an antigen, initiates mucosal immune responses, these findings may shed light on the mucosal adjuvanticity of cationic polymers, such as cationic liposomes.

## 5. Conclusions

In conclusion, we offer a novel insight into mechanisms of how cationic liposomes enhance antigen-specific antibody responses in both mucosal and systemic compartments when nasally immunized with an antigenic protein. Although further experiments will be needed to clarify the whole mechanism(s) of the mucosal adjuvant effects of cationic liposomes, we believe that this mucosal adjuvant system for nasal vaccines may be useful a useful step toward the development of safe and efficient nasal vaccine systems to prevent infectious disease.

## Figures and Tables

**Figure 1 vaccines-08-00008-f001:**
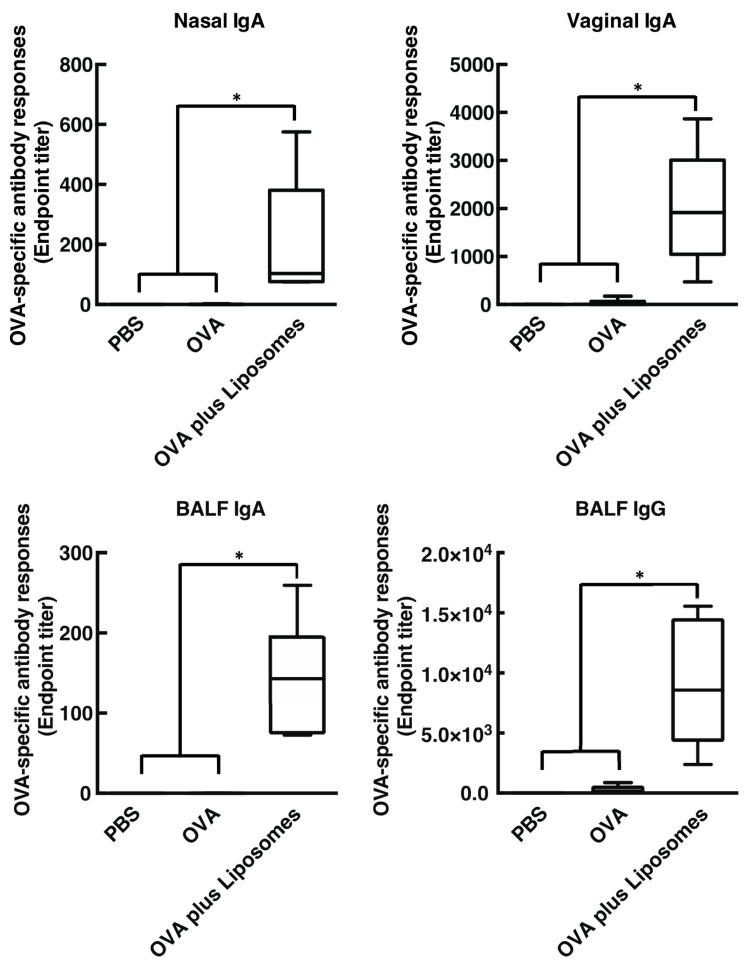
Cationic liposomes stimulate antigen-specific mucosal antibody responses. BALB/cCrSlc female mice were immunized intranasally (days 0, 7, and 14) with phosphate-buffered saline (PBS), egg white ovalbumin (OVA) (2.5 µg/mouse) alone, or OVA (2.5 µg/mouse) together with 1,2-dioleoyl-3-trimethylammonium-propane (DOTAP) and 3β-[N-(N′,N′-dimethylaminoethane)-carbamoyl] (DC-chol) (DOTAP/DC-chol liposomes) (400 nmol/mouse). OVA-specific nasal IgA, vaginal IgA, bronchoalveolar lavage fluid (BALF) IgA, and BALF IgG titers at day 21 were determined by an enzyme-linked immunosorbent assay (ELISA) assay. Data were acquired from at least three biologically independent experiments. The box-plot represents median values with 25th–75th percentiles with error bars indicating 5th–95th percentiles. Significant differences were calculated by Kruskal–Wallis with Dunn’s post-hoc test. * *p* < 0.05.

**Figure 2 vaccines-08-00008-f002:**
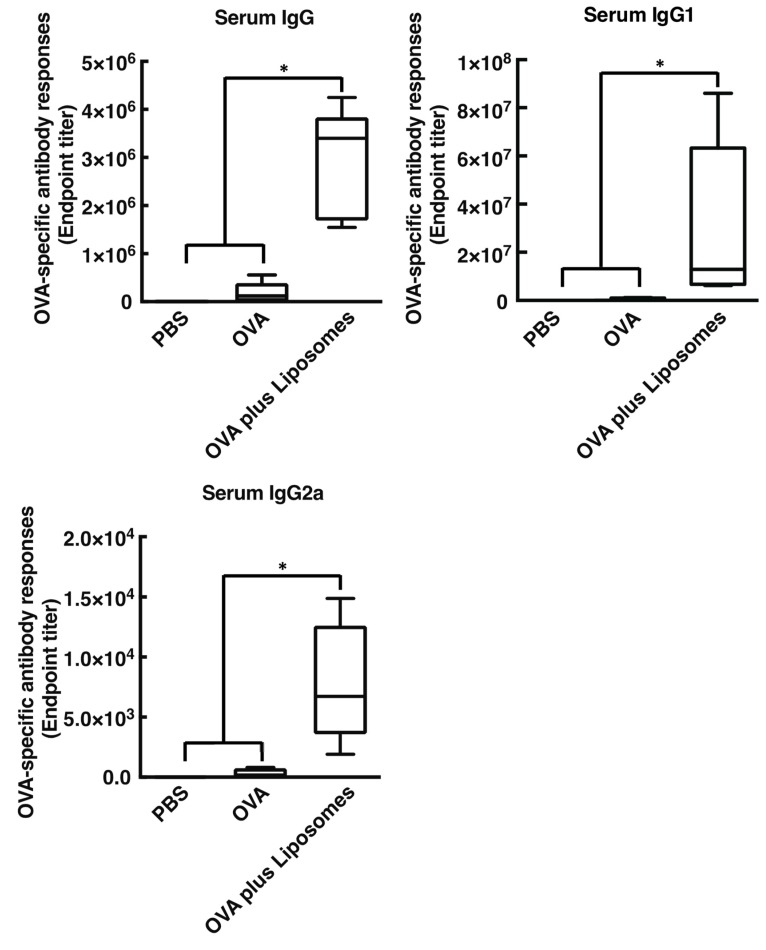
The cationic liposomes stimulate antigen-specific systemic antibody responses. BALB/cCrSlc female mice were immunized intranasally (days 0, 7, and 14) with PBS, OVA (2.5 µg/mouse) alone, or OVA (2.5 µg/mouse) together with DOTAP/DC-chol liposomes (400 nmol/mouse). OVA-specific serum IgG, serum IgG1, and serum IgG2a titers at day 21 were determined by an ELISA assay. Data were acquired from at least three biologically independent experiments. The box-plot represents median values with 25th–75th percentiles with error bars indicating 5th–95th percentiles. Significant differences were calculated by Kruskal–Wallis with Dunn’s post-hoc test. * *p* < 0.05.

**Figure 3 vaccines-08-00008-f003:**
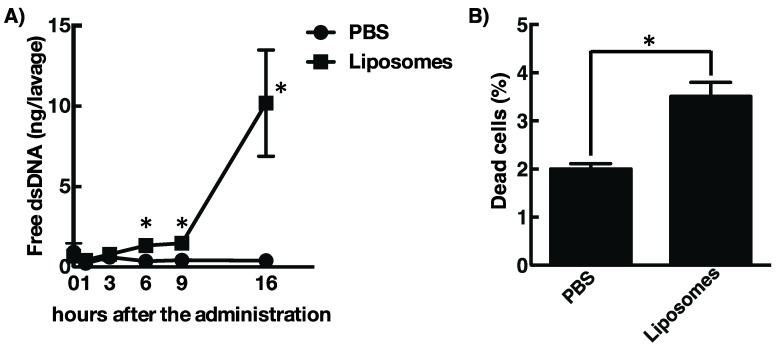
Cationic liposomes induce cell death at the site of administration resulting in the release of host dsDNA. (**A**) The amount of free dsDNA in nasal washes collected from mice i.n. administration with DOTAP/DC-chol liposomes estimated by a fluorescent based method using a Hoechst 33258. (**B**) Rate of dying cells in the nasal mucosa of mice i.n. administration with DOTAP/DC-chol liposomes were assessed by staining with 7-amino-actinomycin D (7-AAD) followed by flow cytometric analysis. At least three independent experiments were conducted, and data are expressed as means ± standard errors. Significance was assessed with *t*-test with Welch correction. * *p* < 0.05.

**Figure 4 vaccines-08-00008-f004:**
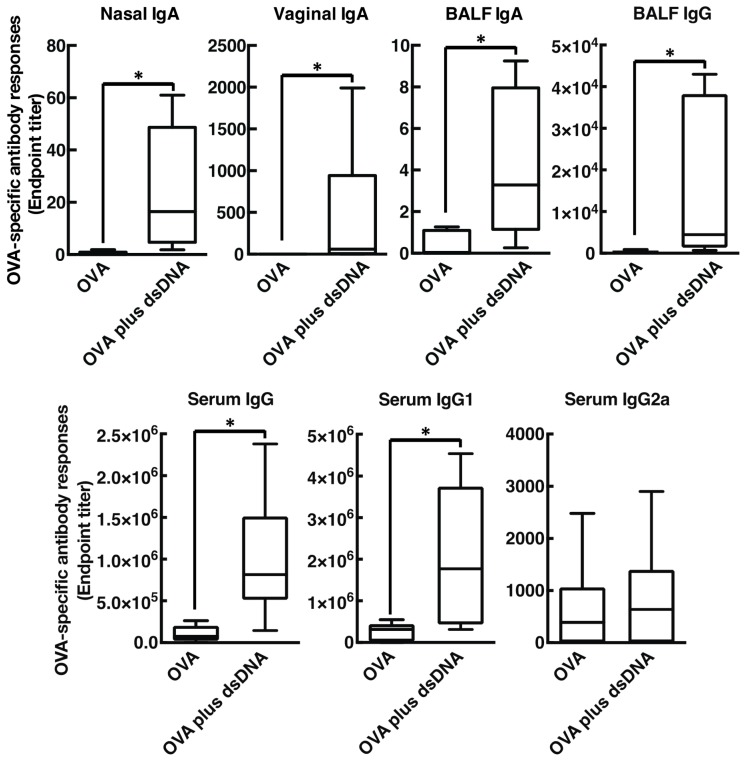
Host dsDNA prepared from BALB/cCrSlc mice potentiate antigen-specific mucosal and systemic antibody responses. BALB/cCrSlc female mice were immunized intranasally (days 0, 7, and 14) with PBS, OVA (2.5 µg/mouse) alone, or OVA (2.5 µg/mouse) together with dsDNA (50 ng/mouse). OVA-specific nasal IgA, vaginal IgA, BALF IgA, BALF IgG, serum IgG, serum IgG1, and serum IgG2a titers at day 21 were determined by an ELISA assay. The data were acquired from at least three biologically independent experiments. The box-plot represents median values with 25th–75th percentiles with error bars indicating 5th–95th percentiles. Significant difference was calculated by Mann–Whitney U test. * *p* < 0.05.

**Figure 5 vaccines-08-00008-f005:**
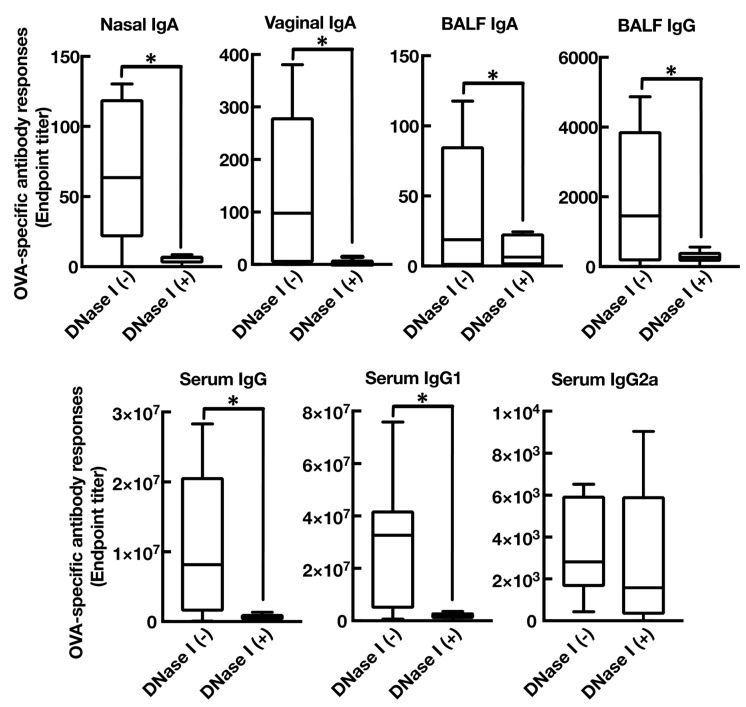
DNase I treatment diminishes antigen-specific antibody responses in BALB/cCrSlc mice immunized with OVA in combination with dsDNA. BALB/cCrSlc female mice were immunized intranasally (days 0, 7, and 14) with OVA (2.5 µg/mouse) together with dsDNA (50 ng/mouse) or OVA (2.5 µg/mouse) together with dsDNA (50 ng/mouse), followed by immediate injection of DNase I (200 IU) nasally. OVA-specific nasal IgA, vaginal IgA, BALF IgA, BALF IgG, serum IgG, serum IgG1, and serum IgG2a titers at day 21 were determined by an ELISA assay. The data were acquired from at least three biologically independent experiments. The box-plot represents median values with 25th–75th percentiles with error bars indicating 5th–95th percentiles. Significant difference was calculated by Mann–Whitney U test: * *p* < 0.05.

**Figure 6 vaccines-08-00008-f006:**
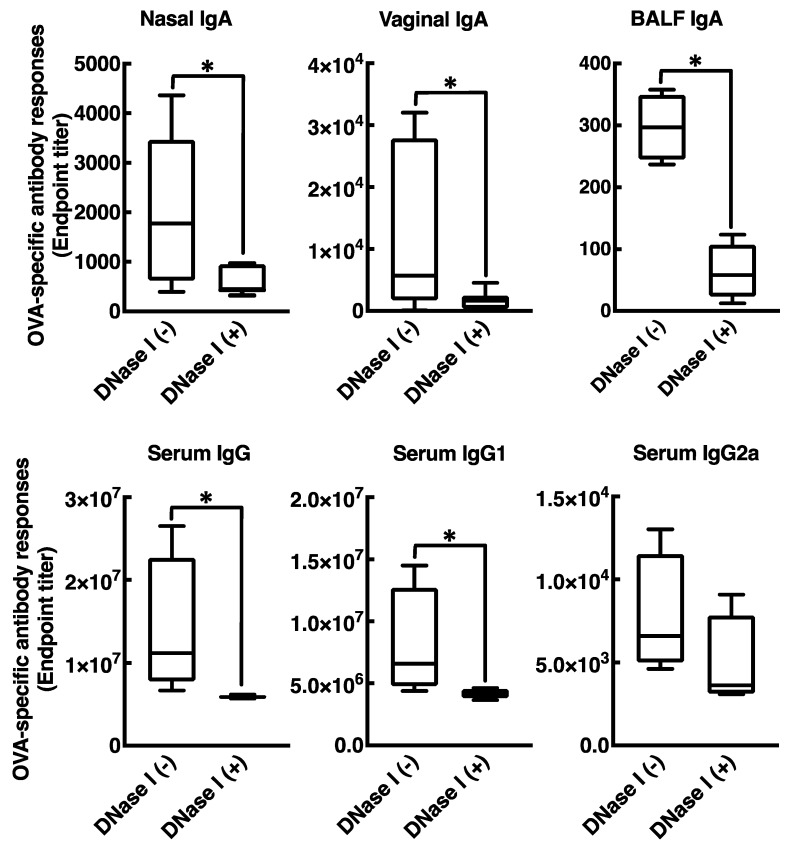
DNase I treatment attenuates antigen-specific antibody responses in BALB/cCrSlc mice immunized with OVA in combination with DOTAP/DC-chol liposomes. BALB/cCrSlc female mice were immunized intranasally (days 0, 7, and 14) with PBS, OVA (2.5 µg/mouse) alone, or OVA (2.5 µg/mouse) together with DOTAP/DC-chol liposomes (400 nmol/mouse). Three and 18 h after each immunization, mice additionally received DNase I (200 IU) nasally. OVA-specific nasal IgA, vaginal IgA, BALF IgA, BALF IgG, serum IgG, serum IgG1, and serum IgG2a titers at day 21 were determined by an ELISA assay. The data were acquired from at least three biologically independent experiments. The box-plot represents median values with 25th–75th percentiles with error bars indicating 5th–95th percentiles. Significant difference was calculated by Mann–Whitney U test. * *p* < 0.05.

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
