# Peer review of "Essential Role of Host Double-Stranded DNA Released from Dying Cells by Cationic Liposomes for Mucosal Adjuvanticity"

_vaccines, 2019, doi:10.3390/vaccines8010008_

Round 1
Reviewer 1 Report
The study by Tada et al provide insights into role of host double-stranded DNA in enhancing adjuvanticity of by cationic liposomes. The results clearly show adjuvant effect of dsDNA and cationic liposomes in antibody responses to mucosal sites. Overall the study is excellent and manuscript is written very well. I have some comments for authors.
Figure 2 : How come the titre of IgG1higher than total IgG. Can you normalize data.
Balf IgA titer : What is Y-axis value? I am not sure if End point titer was so low. Please discuss. It is known that OVA and BSA possess immunologically similar epitopes, a polyclonal antibody produced against OVA will cross-react against the BSA. Therefore, BSA cannot function as a non-relevant blocking for OVA. I am surprised why you didn’t find any cross reactivity. How specific was DNAseI. Is it possible that it has RNase contamination and the adjuvant effect on some degree is also related to RNA. It will be interesting to know the total amount of ova specific Ig rather than just endpoint titer values. In the cell death experiments, did authors check effect of DOTAP and DC-chol alone on cell death. One limitation of this study is no mechanistic link between cationic liposome related enhancement of antibody responses and germinal center response. Since the results indicate dsDNA act as DAMPs and provide adjuvant effect, did the authors plan to study role of TLRs such as TLR9. This will be required to fully comprehend role of dsDNA as adjuvant in future.
Author Response
The authors would like to thank the editor and reviewers for their comments and suggestions that enabled us to improve the quality of our manuscript. All our replies have been prepared to address the reviewers’ comments in a point-by-point fashion.
Comment 1: Figure 2 : How come the titre of IgG1 higher than total IgG. Can you normalize data.
Response:
OVA-specific IgG1 antibody titer appears to be slightly higher than that of the whole IgG titer due to the use of different secondary antibodies for the ELISAs. Additionally, in the present study, we showed anti-OVA antibody responses as endpoint titers, which are not the same as the absolute total amount of Igs. Therefore, it is not appropriate to compare antigen-specific immune responses among different subclasses of the Igs.
Comment 2: Balf IgA titer : What is Y-axis value? I am not sure if End point titer was so low. Please discuss.
Response:
The Y-axis of BALF IgA in the figures is correct and appropriate. This is because antigen-specific IgG titers in BALF are generally much higher than IgA titers in BALF when vaccinated via mucosal routes; additionally, this has been reported by previous studies (Vaccines 2017, 5(3), 19, Acta Biomaterialia 2019, 90, 362, PLOS ONE 2015, 10(5), e0126352).
Comment 3: It is known that OVA and BSA possess immunologically similar epitopes, a polyclonal antibody produced against OVA will cross-react against the BSA. Therefore, BSA cannot function as a non-relevant blocking for OVA. I am surprised why you didn’t find any cross reactivity.
Response:
To the best of our knowledge, rabbit antisera (polyclonal antibodies) produced by the injection of OVA show cross-reaction with BSA. However, no previously published studies have reported about the cross-reactivity of mouse antibodies (normally monoclonal antibodies) against OVA to BSA. Moreover, in most of the previous studies, BSA has been used as a blocking reagent for ELISA to detect anti-OVA antibodies (Biomaterials, 206 (2019) 25-40, Cell, 174 (2018) 1059-1073, etc). Taken together, our protocol for ELISA used in this study is quite appropriate.
Comment 4: How specific was DNAseI. Is it possible that it has RNase contamination and the adjuvant effect on some degree is also related to RNA.
Response:
We agree with this comment. We have the same concerns as those mentioned by the reviewer. Therefore, we used RNase- and protease-free recombinant DNase I from Merck (Roche) for all our experiments. Accordingly, we have modified the description of DNase I in the materials and methods section (page 4, lines 166 and 171).
Comment 5: It will be interesting to know the total amount of ova specific Ig rather than just endpoint titer values.
Response:
The estimation of total amounts of antigen-specific Ig (i.e., absolute amounts) would be complicated due to the lack of positive standards (anti-OVA Ig solutions with known concentrations) for drawing the standard curve. This is the reason why endpoint titer of antigen-specific antibody responses is normally used in such experiments.
Comment 6: In the cell death experiments, did authors check effect of DOTAP and DC-chol alone on cell death.
Response:
We did not check whether or not DOTAP alone and DC-chol alone induce cell death, because it is difficult to prepare stable liposomes composed of DOTAP alone and DC-chol alone. In addition, we reported that the mucosal adjuvanticity of the cationic liposome composed of the combination of DOTAP and DC-chol was the most active. Therefore, we examined only DOTAP/DC-chol liposomes in this study.
Comment 7: One limitation of this study is no mechanistic link between cationic liposome related enhancement of antibody responses and germinal center response.
Response:
We agree with the comment. The analysis of the mechanistic link between cationic liposome-induced antibody responses and germinal center responses is needed for further understanding how cationic liposome enhances antigen-specific antibody responses. Establishing this link will be the focus of our next study.
Comment 8: Since the results indicate dsDNA act as DAMPs and provide adjuvant effect, did the authors plan to study role of TLRs such as TLR9. This will be required to fully comprehend role of dsDNA as adjuvant in future.
Response:
Yes, we are highly interested in elucidating the involvement of DNA sensors such as TLR9, STING, and cGAS in the mucosal adjuvanticity of cationic liposomes. We hope to investigate it and publish the results in the future.
Reviewer 2 Report
The data looks good and well presented The authors need to add more data. Overall, the results are very preliminary and needs to do more experiments I would recommend the authors to add more immune response experimentsAuthor Response
The authors would like to thank the editor and reviewers for their comments and suggestions that enabled us to improve the quality of our manuscript. All our replies have been prepared to address the reviewers’ comments in a point-by-point fashion.
Comment 1: The data looks good and well presented. The authors need to add more data. Overall, the results are very preliminary and needs to do more experiments. I would recommend the authors to add more immune response experiments.
Response:
Thank you for the helpful comments. We agree that many additional experiments are required to thoroughly understand the underlying molecular mechanisms, such as identifying receptors responsible for the recognition of genomic DNA released by cationic liposomes, as well as elucidating how the released genomic DNA induces the following signaling pathways to achieve both mucosal and systemic antigen-specific immune responses after intranasal administration of cationic liposomes. We have already obtained some preliminary results to partially answer these questions. For instance, the intranasal administration of cationic liposomes causes neutrophil accumulation at the nasal cavity. Interestingly, genomic DNA released by cationic liposomes seems to be responsible for neutrophil accumulation. After investigating the role of neutrophils in the mucosal adjuvanticity of cationic liposomes, we hope to publish our findings in the near future.
Reviewer 3 Report
Dear authors,
You have presented a very elegant study and demonstrate an important mechanism of dsDNA immunogenicity which can be an important step in developing future mucosal vaccination strategies. Please perform an extra spelling check with a program and afterwards by personally reading through once more to prevent 'false autocorrect changes'.
The figures also require some editing; make sure all line art has the same thickness (equalize graph sizes and then choose one consistent thickness for all lines, including axes and SD/SEM error bars. Use bold or normal font consistently in axis legends and other text in the graphs, perhaps bold is better visible, but at least it should be consistently used between all graphs, and also equalize font sizes between graphs. Figure 3A:Please write hours instead of hrs on the x axis; 3B is not displayed fully, the text 'liposomes' on the X axis is compromised, this needs to be resolved. For Y axis number labels from 10.000 and higher it may be worth to consider to consistently use a scientific format (for example 4x104 etc.).
Author Response
The authors would like to thank the editor and reviewers for their comments and suggestions that enabled us to improve the quality of our manuscript. All our replies have been prepared to address the reviewers’ comments in a point-by-point fashion.
Comment 1: Please perform an extra spelling check with a program and afterwards by personally reading through once more to prevent 'false autocorrect changes'.
Response:
We have carefully reviewed the manuscript to correct any spelling errors.
Comment 2: The figures also require some editing; make sure all line art has the same thickness (equalize graph sizes and then choose one consistent thickness for all lines, including axes and SD/SEM error bars. Use bold or normal font consistently in axis legends and other text in the graphs, perhaps bold is better visible, but at least it should be consistently used between all graphs, and also equalize font sizes between graphs.
Response:
Based on your comment, we have revised and improved the quality of all figures.
Comment 3: Figure 3A:Please write hours instead of hrs on the x axis; 3B is not displayed fully, the text 'liposomes' on the X axis is compromised, this needs to be resolved. For Y axis number labels from 10.000 and higher it may be worth to consider to consistently use a scientific format (for example 4x104 etc.).
Response:
We have modified all figures according to your comment.
Round 2
Reviewer 1 Report
I think the authors need to include limitations of study. The authors suggest rabbit but not mice antibody show cross reactivity between OVA and BSA. I dont think The rational is correct and certainly highly controversial. I suggest, author discuss this point as limitation of current work in discussion section.
The same is true for DNaseI, unless commercial soure enzyme is verified it is hard to be certain. Discussing this possibility in discussion will help reseachers following this work.
Author Response
The authors would like to thank the editor and reviewers for their comments and suggestions that enabled us to improve the quality of our manuscript. All our replies have been prepared to address the reviewers’ comments in a point-by-point fashion.
Comment 1: I think the authors need to include limitations of study. The authors suggest rabbit but not mice antibody show cross reactivity between OVA and BSA. I don’t think the rational is correct and certainly highly controversial. I suggest, author discuss this point as limitation of current work in discussion section.
Response:
Accordingly, we have added description in the discussion section (page 11, lines 310–316).
Comment 2: The same is true for DNase I, unless commercial source enzyme is verified it is hard to be certain. Discussing this possibility in discussion will help researchers following this work.
Response:
As per your suggestion, we have added description in the discussion section (page 11, lines 337–342).
Reviewer 2 Report
I agree with the authors reply
Author Response
We would like to thank the reviewer for valuable comments and suggestions that enabled us to improve the quality of our manuscript.
Reviewer 3 Report
Dear authors,
To my opinion you have addressed all previous issues and the manuscript is suitable for publication in current form.
Author Response

(The authors gave the same response as above.)
